# DFT Simulation-Based Design of 1T-MoS_2_ Cathode Hosts for Li-S Batteries and Experimental Evaluation

**DOI:** 10.3390/ijms232415608

**Published:** 2022-12-09

**Authors:** Elaheh Hojaji, Eleftherios I. Andritsos, Zhuangnan Li, Manish Chhowalla, Constantina Lekakou, Qiong Cai

**Affiliations:** 1Department of Mechanical Engineering Sciences, Faculty of Engineering and Physical Sciences, University of Surrey, Guildford GU2 7XH, UK; 2Department of Chemical and Process Engineering, Faculty of Engineering and Physical Sciences, University of Surrey, Guildford GU2 7XH, UK; 3Department of Materials Science and Metallurgy, University of Cambridge, Cambridge CB3 0FS, UK

**Keywords:** Li-S battery, cathode, 1T-MoS_2_, sulphur vacancy, reaction pathway

## Abstract

The main challenge in lithium sulphur (Li-S) batteries is the shuttling of lithium polysulphides (LiPSs) caused by the rapid LiPSs migration to the anode and the slow reaction kinetics in the chain of LiPSs conversion. In this study, we explore 1T-MoS_2_ as a cathode host for Li-S batteries by examining the affinity of 1T-MoS_2_ substrates (pristine 1T-MoS_2_, defected 1T-MoS_2_ with one and two S vacancies) toward LiPSs and their electrocatalytic effects. Density functional theory (DFT) simulations are used to determine the adsorption energy of LiPSs to these substrates, the Gibbs free energy profiles for the reaction chain, and the preferred pathways and activation energies for the slow reaction stage from Li_2_S_4_ to Li_2_S. The obtained information highlights the potential benefit of a combination of 1T-MoS_2_ regions, without or with one and two sulphur vacancies, for an improved Li-S battery performance. The recommendation is implemented in a Li-S battery with areas of pristine 1T-MoS_2_ and some proportion of one and two S vacancies, exhibiting a capacity of 1190 mAh/g at 0.1C, with 97% capacity retention after 60 cycles in a schedule of different C-rates from 0.1C to 2C and back to 0.1C.

## 1. Introduction

The rechargeable lithium-sulphur (Li-S) batteries feature the advantages of high theoretical energy density, low toxicity, and low cost, making them promising candidates as the next-generation energy storage technology. Despite extensive development in materials for the cathode, separator, and electrolyte [1], the migration of lithium polysulphides (LiPSs) to the anode and the slow reaction kinetics are still the key challenges that hinder their commercial applications. The impact of the LiPSs migration and shuttling effect is active material loss, rapid capacity fading, poor cycling stability, and severe anode corrosion [2]. Although a solid electrolyte would not allow any shuttling of LiPSs [3], it would also substantially reduce the diffusion rate of the Li^+^ ions [3,4,5], compared to that of cations in liquid electrolytes [5,6]. Efforts to eliminate the problem of LiPSs migration while using a liquid electrolyte have focussed on trapping the soluble LiPSs in the cathode via physical confinement or attraction to functional groups in the cathode host. Nanostructured cathode hosts of hollow carbon particles [7] and micro/mesoporous carbon [8,9] offer physical confinement, despite their limited sulphur loading and inability of cyclic S_8_ to fill ultra-micropores [10,11], where the redox reactions and Li^+^ ion diffusion are expected to take place on the solid surface [12]. Added functional groups such as -OH, -COOH, and -NH_2_ or N- and O- doped carbon have also shown an increased adsorption energy for the carbonaceous hosts in relation to LiPSs, with the adsorption energy predicted to be in the range of 0.5 to 2 eV via density functional theory (DFT) simulations [13,14,15].

Cathode hosts that both trap LiPSs and also act as electrocatalysts are of the greatest interest for the Li-S batteries: trapping LiPSs slows down their migration, while a fast redox reaction to the next stage in the reaction chain ensures a longer discharge curve to higher capacity. Single atom catalysts (SACs) of TM–N_4_–C (TM = Co, Fe, V, and W) are of this type, showing promise of good electrocatalysts with the DFT-predicted adsorption energies of soluble LiPSs in the range of 1.3 to 4.3 eV [16,17,18,19]. However, a low concentration (1–2 at.%) of these single atoms can be achieved in practice in real materials without particle segregation, which makes it challenging if high loading of single atoms for high overall performance is aimed [20]. Polar hosts, such as transition-metal carbides [21,22], oxides/hydroxides [23,24], sulphides [25,26], selenides [27], nitrides, phosphides [28], and their composites [29,30] instead prove to be more promising to enhance the anchoring capability towards polysulphides intermediates and the kinetics of the LiPSs redox reactions, as they act across the full material surface. Among the polar materials, however, only metal sulphides exhibit enhanced electronic conductivity and catalytic properties at a very high kinetic rate of sulphur electrochemistry during cell operation [31,32,33].

The 2D dichalcogenide MoS_2_, as one of the polar metal sulphide hosts with high electrochemical activity, has recently gained attention to direct the polysulphide conversion reaction to an energetically favourable pathway, leading to significantly enhanced redox kinetics and improved cyclic stability [34,35,36]. Different polytype phase transitions for MoS_2_ have been reported, including 2H (hexagonal) and 1T (triangular) MoS_2_ [37,38,39], of which 1T-MoS_2_ has superior electronic conductivity. From the former category, several hybrid constructions have been suggested such as 2H-MoS_2_/rGO [40], 2H-MoS_2_-graphene nanocomposites [41], sulphur/copolymer/2H-MoS_2_ [42], 2H-MoS2@HCS [43], and 2H-MoS_2_/S prepared by decomposing (NH_4_)_2_MoS_4_ [44]. The remarkable catalytic properties of 1T-MoS_2_ and strong anchoring sites for LiPSs encouraged its employment in the form of nanodots in Li-S batteries, which yielded 721 mAh g_S_^−1^ after 300 cycles, while a low electrolyte/sulphur ratio of 4.6 μL mg^−1^ was used [45]. The reported hybrid constructions on 1T-MoS_2_ have been MXene/1T-MoS_2_ nanohybrids [46], SnO_2_/1T-MoS_2_ nanoarray [35] and three-dimensional graphene/1T-MoS_2_ (3DG/TM) heterostructure [47]. The sulphur deficiency in the structure of 2H-MoS_2_ has also been suggested to significantly enhance polysulphides conversion compared with the plain MoS_2_, as a result of providing more kinetically driving force and formation of more thermodynamically stable structures, such as hybrid constructions of reduced graphene oxide (2H-MoS_2_–x/rGO) [48] and hollow mesoporous carbon (2H-MoS_2_-x/HMC) [32]. Nevertheless, no study has been conducted so far on the role of defected structures of 1T-MoS_2_ as the cathode host compared with the pristine 1T-MoS_2_, necessitating a profound theoretical understanding of the anchoring effects toward LiPSs components and the catalytic effects for the redox reactions.

The literature provides a number of DFT simulations on different MoS_2_-based hosts to boost the anchoring effect of the pristine 2H-MoS_2_ toward LiPSs and therefore reduce its dissolution in the chosen electrolyte [26,47,49,50,51]. The DFT predicted adsorption energies of LiPSs by a pristine 1T-MoS_2_ monolayer were also reported in the range of 0.64 to 2.9 eV, compared with 0.1 to 0.8 eV by a pristine 2H-MoS_2_ monolayer [26]. In a separate study, the calculated adsorption energies of LiPSs adsorbed on a pristine 1T-MoS_2_ monolayer were reported in the range of 1.1 to 1.4 eV, which were significantly higher than those of graphene supported 1T-MoS_2_ and free-standing graphene [47]. Despite different reported values of adsorption energies for pristine 1T-MoS_2_, research is still ongoing for an ideal anchoring material with enhanced electrocatalytic effects, good electrical and ionic conductivity, improving the electrochemical performance of Li–S batteries. To our knowledge, the binding effects of the defected 1T-MoS_2_ structures towards LiPSs have not yet been explored. Additionally, the catalytic activity for the LiPSs conversion reactions on the sulphur-deficient 1T-MoS_2_ structures is not known, although the conversion reaction of LiPSs on 2H-MoS_2_ and 2H-MoS_2_-2S during the discharge process [51], has been studied using the Gibbs free energy profiles.

In this study, we use DFT simulations to design 1T-MoS_2_ hosts for Li-S batteries by assessing a pristine 1T-MoS_2_ substrate, an 1T-MoS_2_-1S (with one sulphur vacancy) and an 1T-MoS_2_-2S (with two sulphur vacancies) in terms of their binding energy with LiPSs, the Gibbs free energy profiles for the full polysulphide reaction chain from S_8_ to Li_2_S to identify the rate-limiting step in the conversion reactions, and the reaction pathway from Li_2_S_4_ to Li_2_S, as this section of the pathway is considered the flattest part of the profile with rate-limiting sections [17]. The TSS (transition state search) task of DFT is employed to predict the activation energy barriers to chemical reactions to determine reaction pathways for the typically slow reaction chain of Li_2_S_4_ to Li_2_S_2_ and then to Li_2_S for the different 1T-MoS_2_ substrates. Furthermore, a Li-S battery is evaluated experimentally with a 1T-MoS_2_ cathode host combining the structures investigated in DFT, then its performance is compared to a typical sulphur cathode with a carbonaceous host.

## 2. Results and Discussion

### 2.1. DFT Simulations

The optimised 1T-MoS_2_ and its defected (MoS_2_-1S and MoS_2_-2S) structures, as well as the optimised configurations of Li_2_S_n_ (n = 1, 2, 4, 6, and 8), are shown in Figure 1. All structures and the average bond lengths are in a good agreement with high-accuracy first-principles calculations reported so far [17,52].

For the adsorption of Li_2_S_n_ (n = 1, 2, 4, 6, and 8) on pristine 1T-MoS_2_ and the defected structures, the vertical distances between two Li atoms to the surfaces were set initially according to the reported data [51] and then further adjusted by the DFT simulations of energy minimisation, see Figure 2. For the investigation of Li_2_S_8_ adsorption, the Li–Li bond was considered perpendicular to the 1T-MoS_2_ (001) structures, such that the Li_2_S_8_ molecule lied flat above the surface, consistent with the reported optimisation results [25,51].

Figure 3 presents the adsorption energies of the LiPSs intermediates (Li_2_S_n_, n = 1, 2, 4, 6, and 8) on the 1T-MoS_2_ substrates in comparison with SACs TM-N_4_-C (TM = Co, Fe, V, and W) and graphene. For Li_2_S, Li_2_S_2_ and, especially, Li_2_S_4_, whose migration to the anode is critical in suppressing the subsequent cascade reactions, the strength of binding energy follows the order of graphene < Co-N_4_-C < Fe-N_4_-C < 1T-MoS_2_ pristine < 1T-MoS_2_-1S < V-N_4_-C < W-N_4_-C < 1T-MoS_2_-2S. The strongest binding affinity amongst all substrates is observed for 1T-MoS_2_-2S, with the binding energies of 5.38, 5.14, and 4.04 eV for Li_2_S, Li_2_S_2_, and Li_2_S_4_, respectively. For Li_2_S_6_ and Li_2_S_8_, although the 1T-MoS_2_ substrates with sulphur vacancies perform better than Co-N_4_-C and Fe-N_4_-C, they do exhibit weaker affinity than V-N_4_-C and W-N_4_-C. The observed stronger adsorption energies of the MoS_2_ substrates with the shorter lithium polysulphide chains indicates gradually reinforced chemical interaction from the lithium ions. This is in contrast with the weakest binding strength of the single atom catalysts with Li_2_S_4_, especially for V-N_4_-C and W-N_4_-C, indicating the chance of Li_2_S_4_ migration to the anode before it has the chance to transform to the precipitating solid Li_2_S_2_ and Li_2_S. Our results on the binding energies for the pristine 1T-MoS_2_ showed an energy range of 0.7–3.3 eV which is in a good agreement with those of Dong et al. [26], but not completely in line with those of He et al. [47]. To the best of our knowledge, we are the first to report the binding energies of LiPSs with the defected 1T-MoS_2_, with one and two sulphur vacancies.

The higher adsorption energy for 1T-MoS_2_ with sulphur vacancies toward LiPSs intermediates, especially the lower order sulphides Li_2_S, Li_2_S_2_, and Li_2_S_4_, compared with 1T-MoS_2_ pristine can be explained based on the charge density. Research showed that the sulphur deficiency in the MoS_2_ structure resulted in larger charge densities of surrounding S atoms and therefore a tighter bond with Li atoms [51]. This would enhance the interaction with the LiPSs intermediates, providing more thermodynamics and kinetic driving force for binding and finally causing an increased stabilisation effect. Overall, Figure 3 illustrates that 1T-MoS_2_-2S (two sulphur vacancies) has better capability in trapping all sulphides than 1T-MoS_2_ and 1T-MoS_2_-1S.

To evaluate the catalytic effect of 1T-MoS_2_ substrates, different disproportionation reaction pathways for the slow conversion of Li_2_S_4_ to Li_2_S were investigated:
(i)a two-step process: Li_2_S_4_ → Li_2_S_2_ + S_2_ followed by Li_2_S_2_ → Li_2_S + S;(ii)a one-step process: Li_2_S_4_ → Li_2_S + S_3_.

Figure 4 depicts the optimised geometry of the adsorbed Li_2_S_2_ (Figure 4A–C) and its dissociated products (Figure 4D–F) on the surface of 1T-MoS_2_ substrates for the last reaction: Li_2_S_2_ → Li_2_S + S of pathway (i) above, as obtained by the DFT simulations. Appendix A also show the optimised geometry of the adsorbed Li_2_S_4_ and its dissociated products on the surface of 1T-MoS_2_ substrates for the first reaction: Li_2_S_4_ → Li_2_S_2_ + S_2_ of pathway (i) and the reaction: Li_2_S_4_ → Li_2_S + S_3_ of pathway (ii), respectively. In the different structures in Figure 4, Appendix A, the separated S atom, S_2_ or S_3_ molecules on 1T-MoS_2_ substrates is either on the top of the MoS_2_ structure (1T-MoS_2_ pristine) or embedded in the MoS_2_ structure (1T-MoS_2_-1S) or in the void space created by the vacancies (1T-MoS_2_-2S).

Figure 5 shows the dissociation, $E_{d}$, and activation, $E_{a}$, energies and atomic structures of reactants and products for the disproportionation reactions of (A) Li_2_S_4_ → Li_2_S_2_ + S_2_, (B) Li_2_S_4_ → Li_2_S + S_3_, and (C) Li_2_S_2_ → Li_2_S + S on pristine 1T-MoS_2_, 1T-MoS_2_-1S, and 1T-MoS_2_-2S. The reaction coordinate and energy change of the transition state (TS) (as found from the transition-state search method (TSS)) and product, with respect to the reactant are further presented, with the calculated $E_{a}$ and $E_{d}$ values. Considering the two pathways (i) and (ii), starting from Li_2_S_4_ to ultimately reaching Li_2_S, Figure 5 illustrates that reaction pathway (ii), Li_2_S_4_ → Li_2_S + S_3_, is the preferred pathway for 1T-MoS_2_ pristine and 1T-MoS_2_-1S cathode hosts, as the activation and dissociation energies of this reaction (which are 1.73 and 0.32 eV, respectively) are lower than those of the reactions of pathway (i) (which are 2.30 and 0.68 eV, respectively). Hence, for these two molybdenum disulphide substrates the preferred pathway could be through direct conversion of Li_2_S_4_ to Li_2_S, by-passing the Li_2_S_2_ formation step. However, for 1T-MoS_2_-2S, reaction pathway (i) is preferred, consisting of a first step of Li_2_S_4_ → Li_2_S_2_ + S_2_ followed by the second step of Li_2_S_2_ → Li_2_S + S, as determined from their lower relative energy change, the activation barrier and dissociation energy values which compared well with those of Li_2_S_4_ → Li_2_S + S_3_. These results mean that the minimum amount of energy that must be provided to the reactant components (Li_2_S_4_) to lead to a chemical reaction for the Li_2_S_4_ → Li_2_S + S_3_ pathway is only met on 1T-MoS_2_ pristine and 1T-MoS_2_-1S cathodes and for Li_2_S_4_ → Li_2_S_2_ + S_2_ pathway is only met on 1T-MoS_2_-2S. Figure 5 further indicates a considerable activation energy difference between 1T-MoS_2_-1S and 1T-MoS_2_-2S throughout all reaction pathways. This can be explained based on the relationship between activation energy of a chemical reaction and its rate. In fact, molecules can only complete the reaction once they have reached the top of the activation energy barrier. Therefore, for the higher barriers, fewer molecules will have enough energy to make it over the barrier at any given moment, leading to a lower reaction rate, and for the lower barriers, more molecules will have enough energy to make it through, resulting in an increased reaction rate. Thereby, in Figure 5, very fast reaction rate for the reaction of Li_2_S_4_ → Li_2_S + S_3_ on 1T-MoS_2_ pristine and 1T-MoS_2_-1S cathodes is predicted but for the reaction of Li_2_S_4_ → Li_2_S_2_ + S_2_, followed by Li_2_S_2_ → Li_2_S + S, on 1T-MoS_2_-2S, a reduced reaction rate is expected.

The main observation from all the results of Figure 5 is the lowest activation and dissociation energies for 1T-MoS_2_-1S compared to other molybdenum disulphide substrates throughout all reactions. This highlights the high catalytic activity of 1T-MoS_2_-1S which results in spontaneous conversion reactions on the surface, with minimum energy requirement to proceed along the discharge of a Li-S battery.

A comparison of the activation energies for the typical last reaction Li_2_S_2_ → Li_2_S + S on 1T-MoS_2_ substrates against our previously published results on the single atom catalysts and graphene [17] again confirms the superior catalytic activity of 1T-MoS_2_-1S against all studied catalysts with the lowest $E_{a}$ value ($E_{a}$ follows the order of graphene (2.73 eV) > Fe-N_4_-C (1.71 eV) > Co-N_4_-C (1.66 eV) > W-N_4_-C (1.10 eV) > V-N_4_-C (1.01 eV) > 1T-MoS2-1S (0.27 eV). Therefore, it is expected that 1T-MoS_2_-1S could promote a fast LiPSs reaction and conversion to the final product.

Figure 6 displays the relative Gibbs free energy profiles for the LiPSs disproportionation reactions on 1T-MoS_2_ substrates against graphene and TM-N_4_-C (TM = Co, Fe, V, and W) SAC materials, where an initial spontaneous exothermic reaction from S_8_ to Li_2_S_8_ is followed by a flatter profile with thermoneutral, or a small degree of exothermic or endothermic trends depending on substrate. It can be observed that the first step of Li_2_S_8_ production from S_8_ reactant is the most spontaneous according to the order: 1T-MoS_2_-2S > 1T-MoS_2_-1S > 1T-MoS_2_ > SACs. This means that the fastest reaction kinetics occurred on the 1T-MoS_2_-2S surface with Gibbs free energy value of −10.14 eV for the first conversion step.

Referring to Figure 3 which demonstrates the adsorption energies of the different 1T-MoS_2_ substrates towards Li_2_S_8_ are reducing, we could speculate that the amount of Li_2_S_8_ quantity produced on 1T-MoS_2_-2S would be prevented from migrating to the anode during discharge and be ready for the next reaction step at the surface of the cathode host. The subsequent steps to Li_2_S_6_ and Li_2_S_4_ are almost thermoneutral for the 1T-MoS_2_ substrates, indicating slower reactions than the first step. Thereafter, the preferred direct pathway is a little endothermic for 1T-MoS_2_ and a little exothermic for 1T-MoS_2_-1S, but the energy required for 1T-MoS_2_ is still less than the energy required for graphene [17], demonstrating the effectiveness of 1T-MoS_2_ and 1T-MoS_2_-1S to reach good conversion to the final product Li_2_S. The 1T-MoS_2_-2S substrate guides the reaction mechanism along a two-step pathway, an almost thermoneutral step of Li_2_S_4_ to Li_2_S_2_ and a last endothermic step to Li_2_S of 1.52 eV, the same as for graphene substrate [17]. The very slow reaction conversion of Li_2_S_2_ to Li_2_S plays a major limiting role in the utilisation of sulphur in Li-S batteries and has been identified as the rate-limiting step for the conversion of LiPSs species. Amongst all three of the investigated 1T-MoS_2_ cathode hosts, 1T MoS_2_-1S offers the best substrate for the Li_2_S_4_ to Li_2_S conversion, while 1T-MoS_2_ without defects offers the best substrate for the Li_2_S_4_ production and 1T-MoS_2_-2S offers the best substrate for the spontaneous Li_2_S_8_ production and adsorption at the surface of cathode host. Taking also into account that 1T-MoS_2_-2S and then 1T-MoS_2_-1S have the highest adsorption energy towards LiPSs (Figure 3), it is clear that a balance of pristine 1T-MoS_2_ and defected with 1S and 2S vacancies would be best to combine the strengths of each 1T-MoS_2_ material and counteract any disadvantages.

Figure 7 displays the atom-projected PDOS plots for 1T-MoS_2_, 1T-MoS_2_-1S, and 1T-MoS_2_-2S structures, for the adsorbed Li_2_S_4_ and the products of the disproportionation reactions throughout the preferred reaction pathways for each substrate in the conversion of Li_2_S_4_ to Li_2_S. The up and down spins states of electrons in the PDOS data before Li_2_S_4_ adsorption and throughout the conversion reactions were symmetrically distributed which reflects the non-magnetic properties of the based molecules even after the conversion reactions. It must be mentioned that here for the sake of simplicity and clarity in the graphs only the up spin states of electrons are presented in Figure 7. For the 1T-MoS_2_, 1T-MoS_2_-1S, and 1T-MoS_2_-2S structures before adsorption of Li_2_S_4_, the hybridized DOS intensities for Mo and S atoms below the Fermi level, where there is a high probability of the electrons occupied-states, confirm the inherent covalent binding between the nearest neighbour atoms, in line with previous observations in the literature [53]. The DOS intensities for Mo above the Fermi level demonstrate its partly metallic character as a transition metal. This can be explained based on the availability of electrons unoccupied-states of Mo for the binding. After adsorption of Li_2_S_4_ on the 1T-MoS_2_ substrates, the DOS intensities near-Fermi level exhibit a sudden increase for both Mo and S atoms, indicating occupation of the available states by the Li_2_S_4_ electrons.

Figure 8 further depicts the orbital projected PDOS plots of d, p, and s orbitals of Mo, S, and Li atoms, respectively, after adsorption of Li_2_S_4_ and for the dissociation reactions for all substrates. The plots reveal that the d states have coupled with the p states and have crossed the Fermi level, demonstrating a strong hybridisation between Mo d and S p orbitals throughout the disproportionation reactions. The electron doping effect of Li can also cause the chemical potentials to be shifted towards the unoccupied d bands, and therefore the composite systems to remain metallic. This confirms that the loss of energy with lithiation is plausible, as was already observed in our binding energy results (see Figure 3).

Figure 9 further presents the atom-projected PDOS plots for the last reaction step Li_2_S_2_ to Li_2_S of the preferred pathway on the 1T-MoS_2_-2S substrate, where no significant horizontal shifts are observed with respect to the Fermi level but the number of probability of states has increased especially for Mo near the Fermi level and for S away from the Fermi level throughtout the conversion reaction.

### 2.2. Experimental Evaluation

Based on the recommendations of the DFT simulations in Section 2.1, the 1T-MoS_2_ material used as cathode host combines 1T-MoS_2_, 1T-MoS_2_-1S, and 1T-MoS_2_-2S regions, amounting to a vacancy concentration of 6 × 10^13^ cm^−2^ of which 87% is 1S vacancies and 13% is 2S vacancies according to the data from material characterisation via high angle annular dark-field (HAADF) scanning transmission electron microscope (STEM) imaging from a previous study by our group for hydrogen evolution reaction [54]. Figure 10 presents the experimental data from the electrochemical testing. Figure 10A depicts a maximum specific capacity of 1190 mAh g_S_^−1^ and 655 mAh g_S_^−1^ upon discharge at 0.1C and 1C, respectively, which is superior than Li-S cells in the literature with carbon cathode hosts and the same separator and electrolyte tested in galvanostatic charge-discharge (GCD): for example, a Li-S cell with carbon nanofibre host exhibited 930 mAh g_S_^−1^ and 556 mAh g_S_^−1^ upon discharge at 0.1C and 1C, respectively [55]. The reaction plateau at 2.1 V is corresponding to the conversion of Li_2_S_4_ into Li_2_S, described as the following equation (considering S_8_ as the starting reactant in discharge): 2Li_2_S_4_ + 12Li^+^ + 12e^−^ ↔ 8Li_2_S, consistent with the DFT simulations that predict this to be the preferred pathway for cathode host pristine 1T-MoS_2_ and 1T-MoS_2_-1S.

CV at different rates in Figure 10B displays a distinct peak upon charge and two distinct peaks upon discharge at the typical voltage positions as expected in Li-S cells [55,56]. The CV rate dependence of the overpotential and intensity of CV peaks is as expected in battery cells [57]. The polarisation at 0.2 mV s^−1^ is lower than expected, compared to the CV curve at 0.1 mV s^−1^, because the faster scan rate of 0.2 mV s^−1^ leaves shorter time for reaction and thus the reaction is less sufficient or it might result to a large amount of sulphide production over the solution saturation point that leads to early precipitation that might block good electron transport for further reaction. The cell demonstrates good cyclability in Figure 10A,C (97% capacity retention after 50 cycles in the schedule of Figure 10C) and almost 100% coulombic efficiency in Figure 10D, all attributed to the elimination of the shuttling of polysulphides given the high adsorption energy and good electrocatalytic properties for the related cathode host material as predicted by the DFT simulations. The fall of capacity of the cell cycled at 1C in Figure 10D, from 625 to 460 mAh g_S_^−1^ after 200 cycles, is attributed to issues of solid precipitation (sulphur precipitation in charge or Li_2_S precipitation in discharge) at the moderately high C-rate, rather than any shuttling effects, given that the capacity is stabilised at 460 mAh g_S_^−1^ after 200 cycles till a total 500 cycles tested so far.

The 1T-MoS_2_ cathode host material of this study, combined with 1 and 2 sulphur vacancies, is a significant advance compared with relevant sulphide-based cathode hosts in the literature. For example, N-doped MoS_2_ based on 1T/2H mixed phase MoS_2_ [58] exhibited similarly good cyclability as displayed in Figure 10C in this study but lower discharge capacity (1030 mAh g_S_^−1^) at 0.1 C than our results in Figure 10A. A mesoporous carbon/ZnS/CuS nanocomposite cathode [59] reached a high specific capacity of 1457 mAh g_S_^−1^ at first discharge at 0.1 C but not so good cyclability after a schedule of 50 cycles at different C-rates (1000 mAh g_S_^−1^ at 0.1 C in cycle 45) compared to our results in Figure 10C.

## 3. Methods and Materials 

### 3.1. Computational Method

The simulation framework which takes into account the atomic and electronic properties of 1T-MoS_2_ (including pristine, defected with sulphur vacancies of 1T-MoS_2_-1S and 1T-MoS_2_-2S) was first implemented in VESTA software. The lattice parameters were optimised by allowing the supercell lattice vectors and ionic positions to change in a simulation box with the vacuum space of 35 Å in the Z-direction. A (4 × 4) supercell of MoS_2_ monolayer consisting of 48 atoms, which contains 16 Mo and 32 S, was made up of the primitive cell of MoS_2_. The supercell size was selected so that the 1S or 2S vacancy concentration in 1T-MoS_2_-1S and 1T-MoS_2_-2S, respectively, matched the total vacancy concentration of the material used in the experimental section of this study, reported as 6 × 10^13^ cm^−2^ in Section 2.2. CASTEP was then employed to perform all spin-polarised DFT simulations for the interaction between the lithium sulphide species and each MoS_2_ substrate [60]. A spin polarisation of 2 was used. The exchange and correlation potential was calculated using the generalised gradient approximation (GGA) exchange correlation function as described by Perdew−Burke− Ernzerhof (PBE) [61], and the Brillouin zone was sampled with the Monkhorst-Pack grid using a 2 × 2 × 1 K-point mesh. The valence electrons identified on a plane-wave basis were used with a cut-off energy of 500 eV to ensure the precision of calculations and the tight convergence criteria (energy and force tolerance of 10^−5^ eV and 10^−4^ eV/A°, respectively). The van der Waals (vdW) dispersion correction as described by Grimme’s empirical method was further considered for all the simulations [62].

### 3.2. Computational Models

The studied 1T-MoS_2_ consisted of one molybdenum sheet sandwiched by two sulphur sheets, forming an S-Mo-S structure, where the weakly interacting layers were held together by van der Waals interactions. The (001) facet as the largest exposed surface [51], was selected for simulation of the catalysts. The adsorption sites were chosen at hexagonal close packed (HCP) and face centred cubic (FCC) positions on all catalysts, as recommended [51]. For the defected 1T-MoS_2_ (001) surface, the model was constructed by deleting one or two Top site S atoms at the centre position (S-deficiency), and therefore, the additional S-deficiency position was further investigated as the adsorption site. The atoms of the catalysts were also relaxed during the optimisation process. The polysulphides modelled were: Li_2_S_x_, x = 1, 2, 4, 6, and 8, which are the typical polysulphides produced in Li-S batteries with electrolyte 1M LiTFSI in DOL/DME solvent mixture 1:1 *v*/*v* [63,64].

Regarding the investigations of the reaction pathway and catalytic effects, a model of LiPS disproportionation reactions was considered for the formation of Li_2_S from Li bulk and S_8,_ as described by Equations (1)–(5):(1)$$S_{8}+2\;Li\to Li_{2}S_{8}\;$$
(2)$$Li_{2}S_{8}\to Li_{2}S_{6}+\frac{1}{4}\;S_{8}\;$$
(3)$$Li_{2}S_{6}\to Li_{2}S_{4}+\frac{1}{4}\;S_{8}\;$$
(4)$$Li_{2}S_{4}\to Li_{2}S_{2}+\frac{1}{4}\;S_{8}\;$$
(5)$$Li_{2}S_{2}\to Li_{2}S+\frac{1}{4}\;S_{8}\;$$

The adsorption, $E_{ads}$, and dissociation, $E_{diss\;},$ energies were calculated for the fully relaxed structures as:(6)$$E_{ads\;}=-[E_{\;\left(MoS_{2}\;sub+LiPs\right)\;}-\left(E_{MoS_{2}\;sub}+E_{LiPs}\right)]$$
(7)$$E_{diss\;}=E_{\left(MoS_{2}\;sub+\mathrm{dissosiated}\;LiPs\right)\;}-E_{\left(MoS_{2}\;sub+LiPs\right)}$$
where $E_{\;\left({\mathrm{MoS}}_{2}\;\mathrm{sub}\;+\mathrm{LiPs}\right)\;}$ is the total energy of the adsorbed polysulfide on 1T-MoS_2_ substrate, $E_{{\mathrm{MoS}}_{2}\;\mathrm{sub}}$ is the individual energy of 1T-MoS_2_ substrate, $E_{\mathrm{LiPs}}$ is the energy of an isolated LiPs molecules in a cubic lattice with a cell length of 30 Å, and $E_{\left({\mathrm{MoS}}_{2}\;\mathrm{sub}+\mathrm{dissosiated}\;\mathrm{LiPs}\right)}$ is the total energy of the dissociated polysulfide on 1T-MoS_2_ substrate. The dissociation energy, $E_{diss\;},$ is the required energy to break a bond and form separate molecular fragments. The relative Gibbs free energies were calculated from the total energies of the examined configurations, based on the disproportionation reactions on the cathode host, as described by the general equation as below:(8)$$\;Li_{2}S_{n}\to \frac{m}{8}\;S_{8}+Li_{2}S_{n-m}$$

The activation energy was additionally calculated from the results of the transition state search (TSS) task of DFT simulation based on the transition states between the ${\mathrm{Li}}_{2}S_{n}$ (i.e., reactant) and decomposed ${\mathrm{Li}}_{2}S_{n}$ (i.e., product). The complete linear synchronous transit and quadratic synchronous transit (LST/QST) methods were used for the TSS task, where the root mean square forces per atom convergence criterion was set to be 10^−3^ eV/Å. The LST method performed a single interpolation to a maximum energy and the QST method alternated searches for an energy maximum with constrained minimisations in order to refine the transition state to a high degree, where their combination interpolated a reaction pathway to find a transition state.

### 3.3. Materials and Experimental Methods

Chemically exfoliated 1T-MoS_2_ as described in [65] was used as a cathode host in Li-S battery cells that were assembled in an argon-filled glove box. Specifically, the cathode of 1T MoS_2_-sulphur composite (60 wt% sulphur) was prepared by vacuum filtrating the dispersion of 20 mg 1T-MoS_2_ and 30 mg sulphur powders in carbon disulphide solution (5.0 M in THF) over an anodic aluminium oxide membrane (0.02 µm pore size), followed by drying at room temperature. The areal sulphur loading was 2 mg cm^−2^. The prepared 1T MoS_2_-sullfur composite (60 wt% sulphur) cathode, carbon-coated aluminium foil current collector, Li foil anode, and Celgard 2400 separator were assembled into CR2032 coin cells. The electrolyte was composed of 1.0 M LiTFSI with 0.2 M LiNO_3_ dissolved in DOL/DME (*v*/*v* = 1:1) solvents. The electrolyte/sulphur (E/S) ratio was controlled as 15 μL/mg S. The assembled Li–S coin cells were galvanostatically cycled within the potential window of 1.7–2.8 V on Land battery cycler at various C rates (1C = 1672 mAh/mgS). CV testing was also performed within the potential window of 1.5–3.0 V on biologic VSP-300 potentiostat.

## 4. Conclusions

In this study, first principle DFT simulations were carried out to design 1T-MoS_2_ cathode hosts for Li-S batteries, where the effect of S vacancies was evaluated by investigating pristine 1T-MoS_2_, 1T-MoS_2_-1S, and 1T-MoS_2_-2S. The investigations focussed on the adsorption properties of the 1T-MoS_2_ host materials with respect to the polysulphides Li_2_S_x_; x = 1, 2, 4, 6, and 8; the change of the Gibbs free energy along the reaction steps in the reaction chain to convert S_8_ to Li_2_S, and the electrocatalytic properties of the MoS_2_ host materials and identification of the preferred pathway for the second reaction stage to convert Li_2_S_4_ to Li_2_S that is typically slow in Li-S batteries. The results were also compared to SACs TM-N_4_-C (TM = Co, Fe, V, and W) and graphene cathode hosts [17]. It was observed that the binding affinities of 1T-MoS_2_ substrates toward polysulphides with the shorter lithium polysulphide chains experienced a gradual strength, where 1T-MoS_2_-2S exhibited significant adsorption ability for all LiPSs intermediates among the examined cathode hosts.

The Gibbs free energy profiles indicated that all 1T-MoS_2_ substrates yielded more spontaneous production of Li_2_S_8_ than the SACs and graphene, with 1T-MoS_2_-2S and 1T-MoS_2_-1S exhibiting twice and four times, respectively, lower energy than that of SACs. Given the relatively high adsorption energy of S-vacancy containing 1T-MoS_2_ materials (3.7 and 2 eV, respectively), we believe these hosts will be able to retain the Li_2_S_8_ molecules in the cathode to proceed with the next reaction steps. The Gibbs energy profiles for the next two reaction steps to Li_2_S_6_ and Li_2_S_4_ showed almost thermoneutral processes for the 1T-MoS_2_ substrates with vacancies and a little exothermic reaction for the pristine 1T-MoS_2_, facilitating conversion to Li_2_S_4_ which is well adsorbed by all substrates. Thereafter, direct conversion from Li_2_S_4_ to Li_2_S (solid precipitate) is the preferred pathway for 1T-MoS_2_ and 1T-MoS_2_-1S, with the reaction being a little exothermic on 1T-MoS_2_-1S, and low activation energies of 0.47 eV and 0.32 eV on 1T-MoS_2_ and 1T-MoS_2_-1S, respectively. The preferred conversion pathway on 1T-MoS_2_-2S includes two steps: a thermoneutral reaction from Li_2_S_4_ to Li_2_S_2_ with an activation energy of 3.89 eV and an endothermic reaction from Li_2_S_2_ to Li_2_S (solid precipitate) with an activation energy of 2.41 eV. The intermediate product Li_2_S_2_ is highly retained on 1T-MoS_2_-2S with an adsorption energy of 5.1 eV. The conclusion was that a combination of 1S and 2S vacancies and pristine 1T-MoS_2_ would best benefit from the specific strengths of each type of 1T-MoS_2_. The following experimental evaluation of a Li-S battery cell with the 1T-MoS_2_ cathode through electrochemical testing depicted very good discharge specific capacity of 1190 mAh g_S_^−1^ at 0.1 C-rate, 100% coulombic efficiency, and 97% capacity retention after 60 cycles in a schedule of different C-rates from 0.1 C to 2 C and back to 0.1 C. This successfully confirmed the results and design of the DFT investigation in proving that migration of the soluble polysulphides to the anode and the “shuttling” effect have been eliminated.

## Figures and Tables

**Figure 1 ijms-23-15608-f001:**
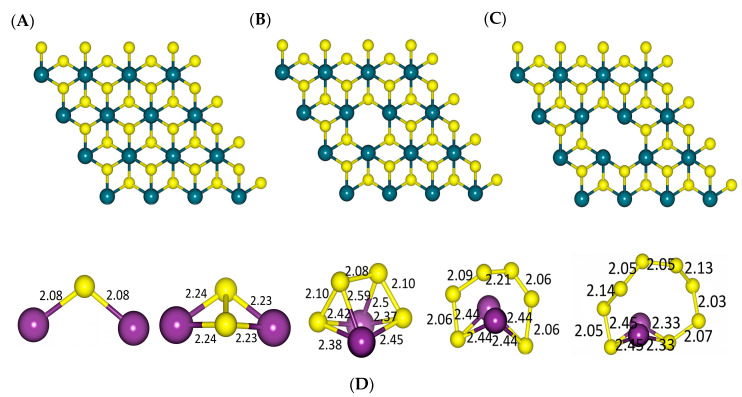
The optimized structures of (**A**) 1T-MoS_2_, (**B**) 1T-MoS_2_-1S, and (**C**) 1T-MoS_2_-2S and (**D**) Li_2_S_n_ (n = 1, 2, 4, 6, and 8), where teal, violet, and yellow colours denote Mo, Li, and S atoms, respectively. Atom bond lengths are in Å.

**Figure 2 ijms-23-15608-f002:**
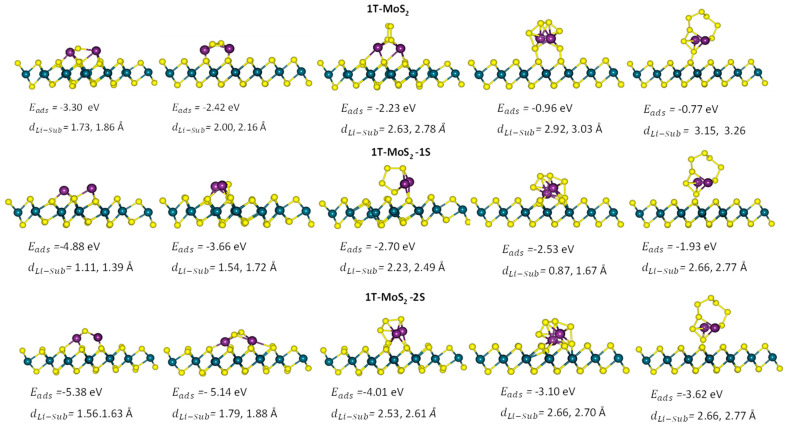
Side view of the optimised configurations for the examined LiPSs on the pristine 1T-MoS_2_ and the defected structures. From top to bottom are pristine 1T-MoS_2_, 1T-MoS_2_-1S, and 1T-MoS_2_-2S and from right to left are Li_2_S_8_, Li_2_S_6_, Li_2_S_4_, Li_2_S_2_, and Li_2_S. $E_{ads}$ and $d_{Li-sub}$ are the adsorption energy of the system and the minimum Li-substrate distance, respectively.

**Figure 3 ijms-23-15608-f003:**
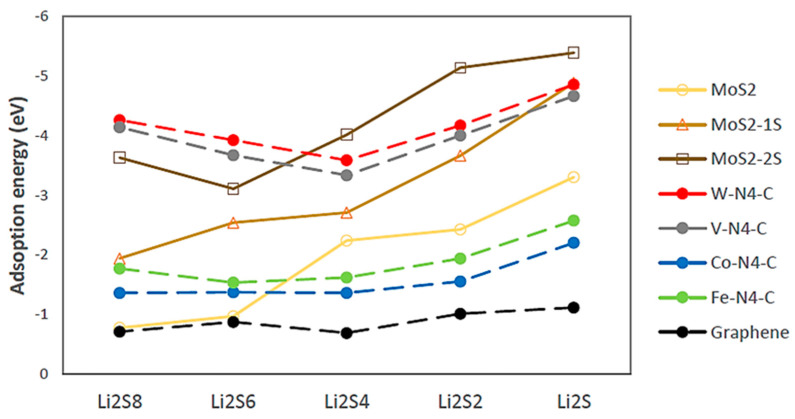
Adsorption energies for Li_2_S_n_ (n = 1, 2, 4, 6, and 8) on 1T-MoS_2_ substrates in comparison with SACs TM-N_4_-C (TM = Co, Fe, V, and W) and graphene [17].

**Figure 4 ijms-23-15608-f004:**
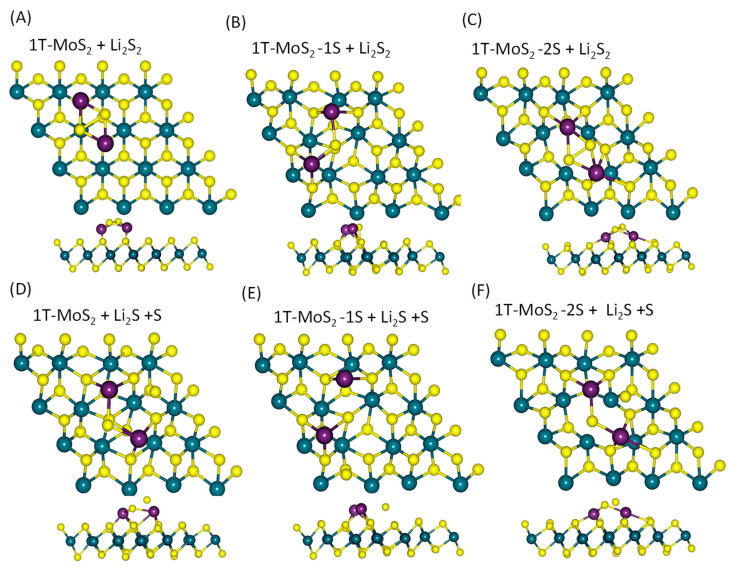
Top views and side views of the optimised geometry of adsorbed Li_2_S_2_ on (**A**) 1T-MoS_2_, (**B**) 1T-MoS_2_-1S, (**C**) 1T-MoS_2_-2S; (**D**–**F**) its dissociated products on 1T-MoS_2_, 1T-MoS_2_-1S, 1T-MoS_2_-2S, respectively. The S–S bond length was less than 5 Å.

**Figure 5 ijms-23-15608-f005:**
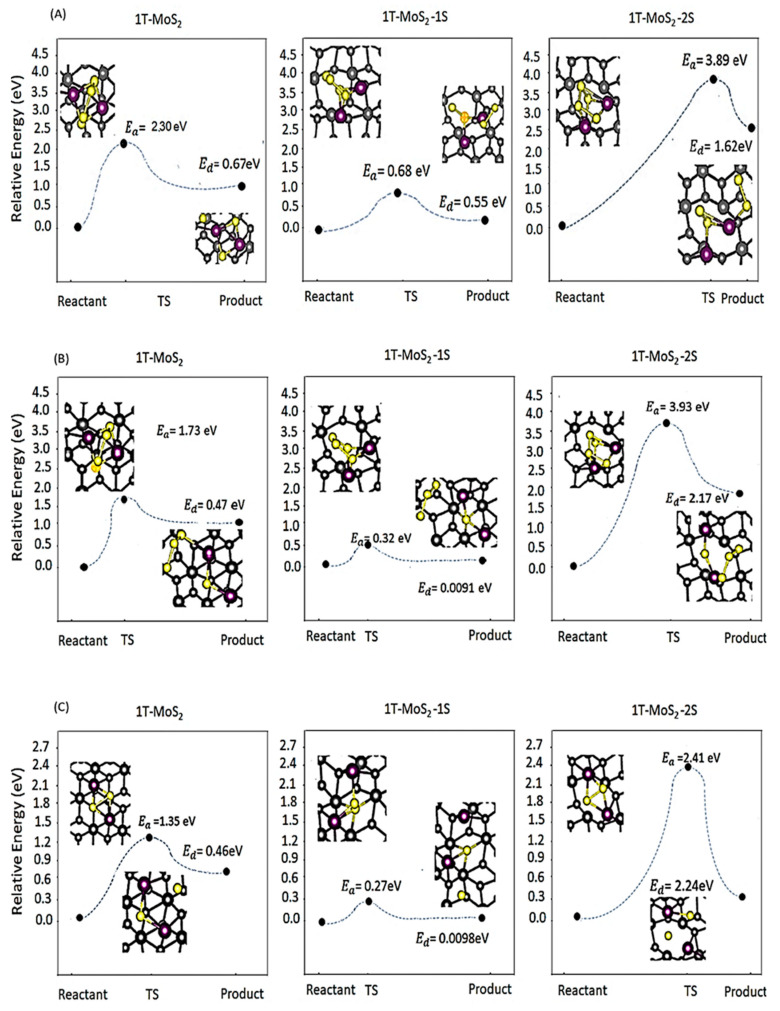
Dissociation energy graphs and atomic structures of reactants and products for the disproportionation reactions of (**A**) Li_2_S_4_ → Li_2_S_2_ + S_2_, (**B**) Li_2_S_4_ → Li_2_S + S_3_, and (**C**) Li_2_S_2_ → Li_2_S + S on 1T-MoS_2_ pristine, 1T-MoS_2_-1S and 1T-MoS_2_-2S. The relative energy change and reaction coordinate of the transition state (TS) and product with respect to the reactant is presented in the graphs. $E_{a}$ is the calculated activation barrier for the transition state and $E_{d}$ is the calculated dissociation energy to break the reactants bonds.

**Figure 6 ijms-23-15608-f006:**
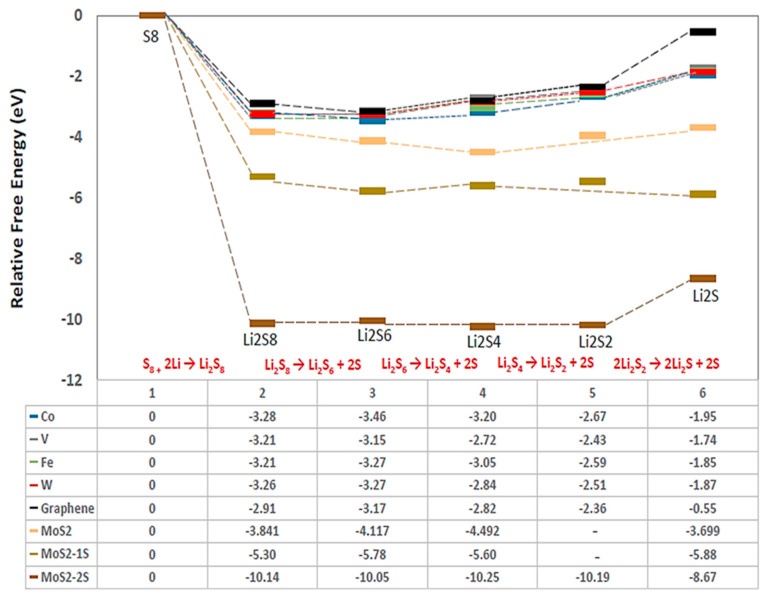
Relative Gibbs free energy for the examined disproportionation reactions on 1T-MoS_2_ substrates in comparison with graphene and TM-N_4_-C (TM = Co, Fe, V, and W) materials. The numerical values show the stepwise relative free-energy change of each reaction. The blank for MoS_2_ plain and MoS_2_-1S represents the direct conversion reaction from Li_2_S_4_ to Li_2_S according to the preferred pathway for these substrates.

**Figure 7 ijms-23-15608-f007:**
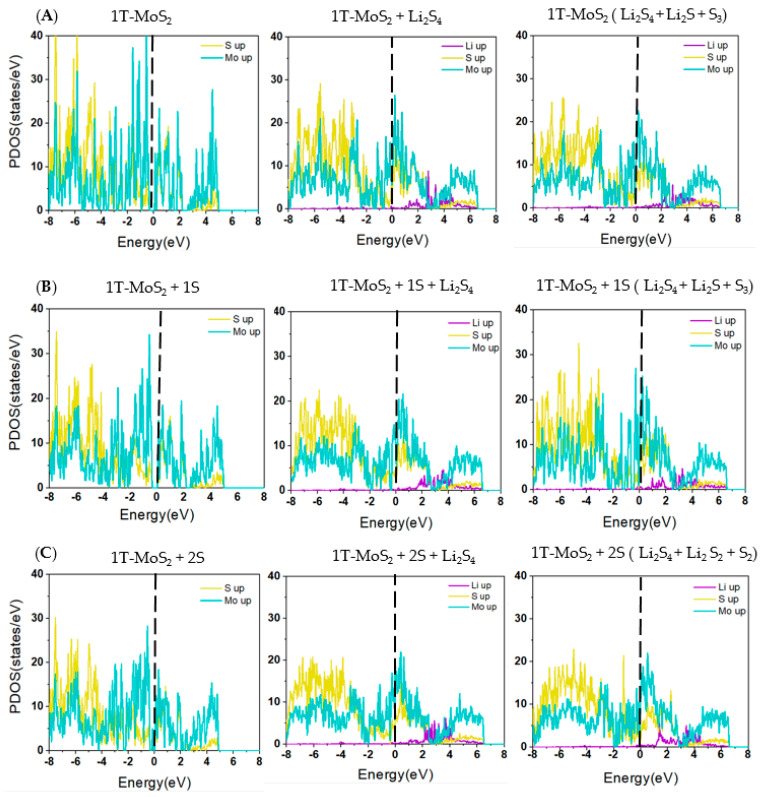
Spin-resolved projected DOS profiles for (**A**) 1T-MoS_2_ pristine, (**B**) 1T-MoS_2_-1S, and (**C**) 1T-MoS_2_-2S for the disproportionation reactions of the preferred pathway for the conversion of Li_2_S_4_ to Li_2_S. The vertical dotted lines represent the Fermi energy level.

**Figure 8 ijms-23-15608-f008:**
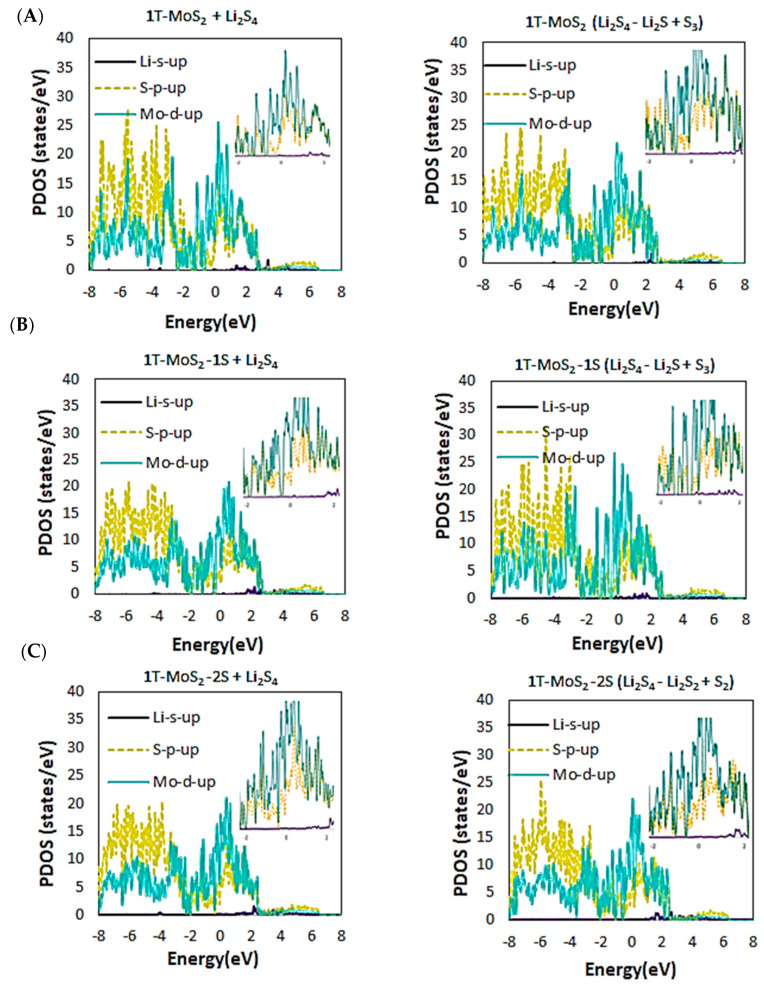
The orbital projected PDOS profiles for the preferred disproportionation reactions on (**A**) 1T-MoS_2_ pristine, (**B**) 1T-MoS_2_-1S, and (**C**) 1T-MoS_2_-2S. The inserts show enlarged intensities around the Fermi energy levels.

**Figure 9 ijms-23-15608-f009:**
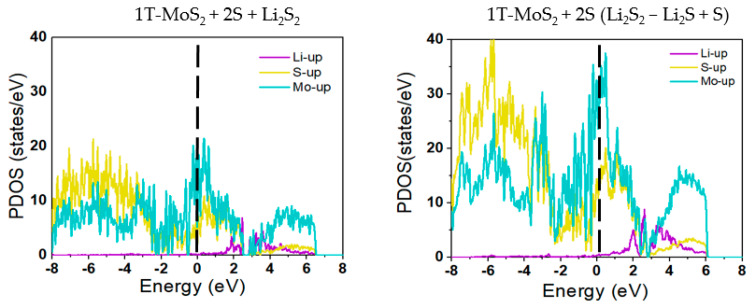
Spin-resolved projected DOS profiles for 1T-MoS_2_-2S for the last disproportionation reaction Li_2_S_2_ to Li_2_S conversion in the two-step preferred pathway for this substrate. The vertical dotted lines represent the Fermi energy level.

**Figure 10 ijms-23-15608-f010:**
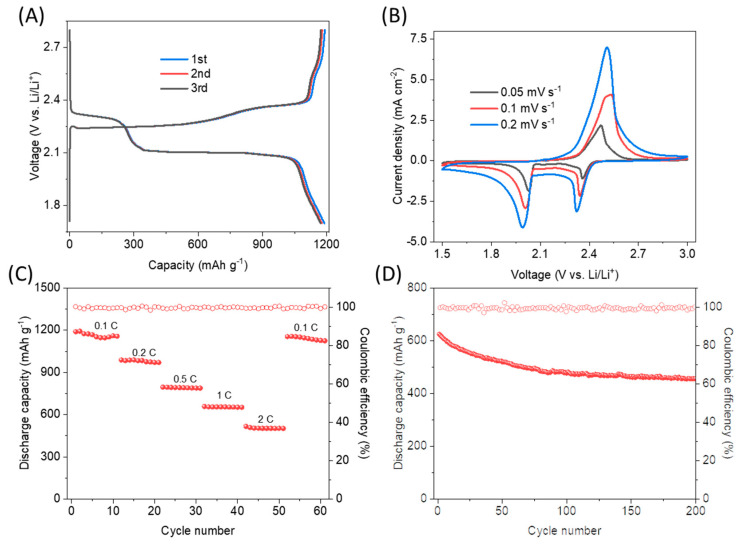
Experimental data from the electrochemical testing of the Li-S cell with the 1T-MoS_2_ cathode that contains a mixture of pristine 1T-MoS_2_, and 1T-MoS_2_-1S, and 1T-MoS_2_-2S: (**A**) The first three GCD (galvanostatic charge-discharge) cycles at 0.1C rate; (**B**) CV tests at different scan rates; (**C**) results of specific capacity (with respect to the sulphur mass) in discharge and Coulombic efficiency during GCD cycling at different C-rates; (**D**) results of specific capacity in discharge and Coulombic efficiency during GCD cycling at 1C.

## Data Availability

Data will be provided after reasonable requests.

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
