# Peer review of "DFT Simulation-Based Design of 1T-MoS_2_ Cathode Hosts for Li-S Batteries and Experimental Evaluation"

_ijms, 2022, doi:10.3390/ijms232415608_

Round 1

Reviewer 1 Report

The manuscript seems to me rather interesting. On the example of MoS2 substrates (pristine and defected) the authors attempted to design the hosts for the Li-S batteries. Using a high-level DFT approach, the authors analyze the set of their electronic characteristics as well as determine the energy characteristics, namely adsorption energy of LiPSs to these substrates, the Gibbs free energy profiles for the reaction chain, and perform the transition state search of the reaction mechanisms.

In my opinion, the manuscript presents solid theoretical research. It does not contain serious drawbacks. It is well written and provides interesting results. The authors adequately select the level of theory and modeling technique. The authors have detailed everything so that interested parties can repeat the presented procedures. The results obtained may be of interest for further experimental studies. Perhaps, the results may indeed be useful in the commercial sector. The novelty of the research is also confirmed.

On the presentation of the results, I have no critical remarks. However, I have a small comment about the choice of the kinetic energy cutoff for wave functions and sampling of k-points. The authors choose the value of 500 eV (~37 Ry) for energy cutoff and a 2x2x1 k-point grid. At first glance, this may not be sufficient. Did the Authors make the confirming convergence tests using increasing cutoff energy? Did the Authors test the convergence at finer grids of k-points? Did the authors remake the calculations with larger supercells? Did the authors check the convergence of energy with changing supercell parameters? I hope the authors will clear up these little technical details. Therefore, I think the manuscript can be accepted for publication in International Journal of Molecular Sciences.

Author Response

Dear Editor

Thank you for the valuable comments and recommendations of the Reviewers. We are submitting a revised paper with all changes in the text tracked red using the “track changes” function in word. More specifically, our point-by-point responses are provided below:

REVIEWER 1

However, I have a small comment about the choice of the kinetic energy cutoff for wave functions and sampling of k-points. The authors choose the value of 500 eV (~37 Ry) for energy cutoff and a 2x2x1 k-point grid. At first glance, this may not be sufficient. Did the Authors make the confirming convergence tests using increasing cutoff energy? Did the Authors test the convergence at finer grids of k-points? Did the authors remake the calculations with larger supercells? Did the authors check the convergence of energy with changing supercell parameters? I hope the authors will clear up these little technical details. Therefore, I think the manuscript can be accepted for publication in International Journal of Molecular Sciences.

Response: Yes, we conducted confirming convergence tests by playing around at different cut-off energies like 480, 520 and 540 and also convergence tests at finer grids of 3x3x3, 4x4x1, 5x5x1 and 6x6x1, with convergence problems for the finer grids. A larger supercell size of (6x6) did not improve the convergence in the problematic grids. The only thing that helped in the convergence of these systems was altering the spin polarisation number, where Spin: 2 worked very well. This detail of spin polarisation number has been added in section 3.1 of the revised manuscript.

Yours sincerely

Constantina Lekakou and Co-authors

Reviewer 2 Report

In this paper the authors provided detailed computational and experimental study of the 1T-MoS2 design for Li-S battery application. By carrying out DFT calculations it was found that the defected MoS2 with S vacancy showed enhanced reactivity and the vacancy concentrations were mapped to real experimental conditions. The reviewer is not an experimental expert thus can only comment on the computational aspect. Overall, the paper is clearly written and has a great contribution to the field, however there are some minor issues that need to be addressed before being accepted:

1.     The main takeaway is the “benefit of a combination of 1T-MoS2 regions, without or with one and two sulphur vacancies”, however given that 1T-MoS2-1Vac is way more superior than 1T-MoS2-2Vac, it is not clear why we need 1T-MoS2-2Vac(?) Perhaps the reviewer missed the point but it would be great to highlight this somewhere so that the reader won’t miss.

2.     Are the Li2Sx structures well established (based on experiments)?

3.     In general, all the images in this manuscript need to be in a higher resolution. E.g., in Figure 6 the colors are not distinguishable.

4.     Figure 3: what is “U=3”? If it’s DFT+U it needs to be specified in the caption.

5.     Figure 5: how were the dashed lines drawn? Were they generated from CASTEP directly or from other software? The curves especially in (B) looked quite suspicious to the reviewer.

6.     It would be a good addition to have a simple kinetic model that predicts the overall reactivity of 1T-MoS2-Vac relative to 1T-MoS2 (ki=viexp(-Ea,i/kBT)) and see how it compares with experiments.

7.     Line 133: dli-sub should be dLi-sub

Reviewer 3 Report

Summary:

This Article ijms 2053325 titled, “DFT simulation-based design of 1T MoS2 cathode hosts for Li-S batteries and experimental evaluation,” reports the calculation result and experimental data of 1T-MoS2 as a cathode host for Li-S batteries. The DFT calculation confirms the benefits to adopt 1T-MoS2 in the lithium-sulfur cell. The experimental data show a discharge capacity of 1190 mAh/g at C/10 rate.

General comment:

The research reports the calculation and experimental data. Minor revisions are suggested to provide the necessary data and parameters. Hope the authors feel the comment useful.

Comments:

(1) It is suggested to give the necessary details of the cathode preparation of the 1T MoS2-sullfur composite cathode. The current collector and/or the cathode substrate is suggested to be described. The 1T MoS2-sullfur composite in the cathode is suggested to have it sulfur loading and final sulfur content in the cathode.

[Suggestion] Please give the necessary experimental data

(2) The discharge and charge curves show an additional reaction that reversibly happens at 2.1V. Related discussion is suggested. In the CV analysis, the 0.2 mv s-1 data shows a lower polarization than the cell at 0.1 mv s-1. Why?

[Suggestion] Please check again the cell performance and make the necessary discussion.

(3) The rate performance and cycling performance should be clarified as discharge capacity or charge capacity. The Y axis is suggested to be revised. In the rate performance, the corresponding discharge/charge efficiency is necessary.

[Suggestion] Please revise the data presentation in the rate and cycling performance data.

(4) Relative sulfide based additive lithium-sulfur cells are suggested in the introduction for showing the research trend and highlighting the breakthrough of this research (10.1016/j.cej.2022.135568: Nitrogen-doped MoS2; 10.1016/j.carbon.2022.04.013: ZnS/CuS; 10.1021/acsami.1c18871:P2S5)

[Suggestion] Please update the reference in support the importance of this research.

(5) In the whole manuscript, it is suggested to have the same unit format. Moreover, according to the information given, this research reports the lithium-sulfur cell rather than battery. Please also review and revise the manuscript to remove the errors and missing symbols.

[Suggestion] Please review and proofread the manuscript.
